# Nutritional and Phytochemical Characterization of Radish Leaves: A Comprehensive Overview

**DOI:** 10.3390/foods14183270

**Published:** 2025-09-20

**Authors:** Xiong Geng, Ziyi Gong, Weina Tian, Miaomiao Zhuang, Huayan Shang, Yujie Chen, Jianrong Li, Yanfang Lv, Kaihua Bai

**Affiliations:** 1College of Food Science and Engineering, Bohai University, Jinzhou 121013, China; 18748135846@163.com (X.G.); xiaoxiong314@163.com (Z.G.); kjcy10@163.com (W.T.); zmm13865981106@163.com (M.Z.); shy2023015150@bhu.edu.con (H.S.); chen2025015093@163.com (Y.C.); lijr6491@163.com (J.L.); 2Inner Mongolia Yili Industrial Group Co., Ltd., Hohhot 010110, China

**Keywords:** radish leaves, phytochemistry, pharmacological activities

## Abstract

Radish is a root vegetable that is widely consumed globally. Radish leaves are typically not consumed and regarded as by-products in agricultural, industrial, and domestic settings. Accumulating evidence suggests that radish leaves possess higher nutritional value compared to their roots, primarily due to their elevated levels of protein, ash, dietary fiber, and ascorbic acid. In light of the growing emphasis on waste reduction and value-added utilization, the application of radish by-products has garnered increasing attention. This study comprehensively reviews the phytochemical composition and pharmacological effects of radish leaves, a common agricultural by-product, detailing the structures of isolated compounds and discussing their chemical properties and bioactivities. When classified by their structural characteristics, these compounds encompass carbohydrates, enzymes, flavonoids, glucosinolates, organic acids, phenolic compounds, sulfur compounds, polysaccharides, and other constituents. Key bioactive components exhibit antioxidant properties, acetylcholinesterase inhibitory activity, antitussive effects, along with anticancer, antihypertensive, anti-inflammatory, antimicrobial, anti-obesity, antiulcerative, and intestinal motility stimulation activities. Radish leaf extracts demonstrate significant therapeutic potential across multiple disease areas, particularly in anticancer and antioxidant applications.

## 1. Introduction

Radish (*Raphanus sativus* L.), also known in ancient Chinese as lai tai, hail sudden, and vegetable head, is included in the Tang Materia Medica. It is a common vegetable in the cruciferous radish genus and is a herbaceous plant with a life cycle of 1~2 years [1]. Radish has developed a rich diversity in morphology, bionomics, and genome through long-term evolution in its primary center of origin in the eastern Mediterranean and the Middle East, as well as through domestication and breeding in multiple secondary centers across Europe, Central Asia, Southern Asia, and Eastern Asia [2]. This vegetable is a very important ingredient in many cuisines, especially in Japan, China, and Korea. This is largely due to the extensive crossing and selection of radish, particularly in East Asian countries, which have produced a large number of cultivars with an especially wide variation in root shapes. While the root is most commonly used part of the plant, the fresh leaves of some cultivars are also used. Many kinds of radishes exist. Their main differences are their colors, shapes, and sizes. Every country has its own representative radish. Figure 1 shows the various types of radishes found in different countries around the world.

Radishes are cultivated worldwide, with specific types adapted to different climates and regions such as Europe and Asia. Common varieties include red globe, black, daikon, and watermelon radishes, each with unique characteristics. Small-rooted and short-season types are cultivated mostly in Europe, whereas large-rooted and long-season radishes are cultivated mostly in Asian countries. Other forms, such as oilseed radish and rat-tail radish, are grown in Southeast Asia [3].

China has with largest number of radish species in the world. Some of the more representative radishes include the Wei County radish from Shandong Province, Meimei radish from Beijing, and Northeast carrots, among others [4]. The radish varieties cultivated in China are predominantly characterized by white skin and white flesh, as well as green skin and green flesh. Among a total of 189 recorded varieties, 72 exhibit white skin and white flesh, 50 exhibit green skin and green flesh, and 38 exhibit red skin and white flesh. The remaining types occur in relatively smaller numbers [5]. Chinese radishes not only have a unique flavor but also play an important role in local food culture and medicinal value [6]. Japan also has a wide range of radish varieties, including both traditional characteristic varieties and high-yield improved varieties examples include Shenghuyuan turnip, white jade radish, Japanese red No. 2 radish and different types of carrots [5]. These traditional and improved high-yield radishes demonstrate the diversity and refined cultivation techniques in Japanese vegetable breeding. South Korea is also an important producer and consumer of radishes. Korean kimchi is widely renowned and in South Korea; all kinds of radishes are processed into kimchi for consumption. South Korean radish varieties mainly include white jade spring radish, New Seoul white jade radish, Changchun radish, and Korean long yellow radish [7].

According to the Food and Agriculture Organization of the United Nations (FAO), the global radish harvest area has exhibited a fluctuating downward trend in recent years, decreasing from 1169.4 thousand hectares in 2010 to 1085 thousand hectares in 2018. From 2018 to 2021, the area remained relatively stable at approximately 1100 thousand hectares, with a slight recovery to 1096 thousand hectares in 2021 compared to the previous year. In terms of global radish production, although the harvested area has generally declined, advancements in agricultural technology have led to increased yields per unit area, contributing to a steady upward trend in total production. Statistical data indicate that global radish production rose from 34.964 million metric tons in 2010 to 41.667 million metric tons in 2021. Regarding the geographical distribution of production, China remains the world’s largest radish producer, accounting for 43.6% of global output in 2021. Other significant producers include Uzbekistan, the United States, Russia, and Germany, with annual outputs of 3.156 million, 1.433 million, 1.303 million, and 0.962 million metric tons, respectively. Taking China as an example, a rough estimate shows that the annual production of root and tuber vegetable waste is approximately 390 million tons.

Radishes typically reach a height of 20 to 100 cm. The fleshy roots are generally elongated-oval, spherical, or conical in shape, and exhibit a periderm coloration ranging from green to white or red. The stems are branched, glabrous, and possess a slight powdery bloom. The basal and lower stem leaves of radish are characteristically large and exhibit a pinnately semi-lobed morphology. Radish leaves are clustered on the shortened stem during the growth period, and their morphological characteristics—including shape, size, and color—vary depending on the variety. Appendix A provides diagrams of the roots and leaves of three common types of radishes. The leaf proportion also differs among radish varieties, ranging from 8 to 30 cm in length and 3 to 5 cm in width. The terminal lobe is ovate in shape, while the 4 to 6 pairs of lateral lobes are oblong, bearing blunt teeth and sparsely covered with coarse hairs. The upper leaves are oblong in form, displaying serrated margins or approaching an entire margin [8]. Analysis of radish leaf morphology indicated that, among 110 radish varieties with documented leaf type information, pinnately lobed leaf varieties are more prevalent, accounting for 73 varieties, whereas flat leaf varieties are comparatively fewer, comprising only 37 varieties [5].

Radish leaves are often discarded as waste. Market research has shown that most radish leaves are either discarded in the field during harvest or removed about 20 days before harvest to increase the yield. As a result, the leaves are a common form of agricultural by-product waste. Radish leaves, commonly known as radish tassels or turnip leaves, are sweet, bitter, and flat. Their main function is to aid digestion and regulate qi [9]. Radish leaves are rich in vitamin C, riboflavin, folic acid and other nutrients, as well as zinc, magnesium, iron, calcium and other trace elements. Their effects include detoxification, dissipation of blood stasis, and the elimination of food, phlegm, diuretics, and thirst, among others. Radish leaves are also rich in flavonoids, polyphenols, polysaccharides and other natural active ingredients with antioxidant, hypoglycaemic, hypotensive, gastrointestinal peristalsis promoting effects, and gastric ulcer-treating functions [10,11]. Chemical analysis has revealed that radish leaves primarily consist of essential oils and a range of vitamins, while also containing trace elements and antiviral compounds [9]. According to modern nutritional studies, radish leaves exhibit higher nutritional value than radish roots in several aspects. For instance, the vitamin C content in radish leaves exceeds that in radish roots by 2 times, while the levels of calcium, magnesium, iron, zinc, riboflavin, and folic acid were found to be 3 to 10 times higher in radish leaves compared to radish roots [10].

As shown in the food nutrition content table, the calcium content in radish leaves was found to reach 238 mg, which is the highest of all components. The content of VC in the leaves was 51 mg/100 g, which is twice the content in the roots. The total phenolic content in the leaves was found to be 695.07 mg/100 g, and total flavonoids was 1042.73 mg/100 g [11]. The specific nutrient contents are presented in Table 1. The American Center for Public Science has scored the nutrition of vegetables according to calories, vitamin K, lutein, vitamin C, and other items, and the results show that radish leaves rank third among super vegetables. Radish contains phytochemical components, including flavonoids, nonflavonoid polyphenols, fats and fat-related compounds, terpenes and their derivatives, glucosinolates, and others. These main components are found in higher concentrations in the leaves and buds, which also have higher nutritional value and higher levels of bioactive compounds than the roots [10].

In the Upper Douro region of northern Portugal, turnip leaves are incorporated into numerous traditional dishes and prepared as a local specialty. In southern Portugal, turnip leaves are utilized in soups, rice dishes, and as a side dish for mackerel. In the United States, turnip leaves are a hallmark of Southern cuisine and are employed in broths flavored with small quantities of cured pork or ham hock. In China, turnip leaves are often stir-fried, consumed cold, boiled in soup, and pickled by individuals. Modern pharmacological studies have demonstrated that radish leaves possess the following functions: promotion of gastrointestinal peristalsis, treatment of gastric ulcers, antioxidant properties, lowering of blood pressure, and lowering of blood sugar. Over the past three decades, a plethora of in vitro and in vivo studies have demonstrated that radish leaves possess a plethora of biological activities.

In recent years, scholars at home and abroad have conducted comprehensive research on the utilization of radish leaves. Wu Haiqing et al. made pickles from radish leaves and systematically analyzed the changes in nutritional components and microbial dynamics at different processing stages [12]. Dong Zhouyong et al. used ultrasonic-assisted extraction to separate chlorophyll from radish leaves, with an average extraction rate of 0.413% [13]. Da-Hee Chung et al. demonstrated that an ethyl acetate extract of radish leaves has antioxidant and anticancer properties and showed the potential to regulate blood pressure in spontaneously hypertensive mice [14].

Radish leaves are a widely available resource. With the development of biology, nutrition and immunology, research on their bioactive components has also deepened. These bioactive compounds have been explored for various uses, including as functional ingredients in health products or food formulas with antioxidant, immunomodulatory, and anti-obesity properties. In addition, due to their biological functions, radish leaf extracts may also serve as carriers for anticancer and hypoglycemic drugs.

Out of 500 references, 96 met our inclusion criteria. A rough count shows that over 460 papers have explored the chemical composition, functional properties and application potential of radish leaves, while more than 40 studies have focused on their morphological characteristics. Despite some progress, several challenges remain: (1) research has mainly focused on white radish leaves, with less attention paid to other radish varieties; (2) there are differences in the extraction efficiency of functional components, and most extraction methods have not yet been industrialized; (3) insufficient research on product development limits the ability to meet the growing demands of the health food industry.

Due to the limited research, the overall utilization rate of radish leaf by-products remains very low, leading not only to resource waste but also to environmental pollution. To address this issue, future research should focus on optimizing extraction techniques, clarifying the structural and conformational characteristics of bioactive compounds and evaluating their functional activities. These efforts are expected to help in discovering new bioactive substances, elucidating a structure–activity relationship, and establishing optimal industrial-scale extraction schemes. Such research will provide a solid theoretical and technical foundation for the development of innovative radish leaf functional foods, thereby better aligning product development with consumer demands. In view of the above, this study presents a comprehensive review of the phytochemical and pharmacological effects of radish leaves based on the available research data, with a view to providing a basis and application prospect for further research on radish leaves.

**Table 1 foods-14-03270-t001:** Nutrient content of radish leaves per 100 g [15].

Nutrient	Content	Nutrient	Content	Nutrient	Content
WATER	91.8 g	VE	0.87 mg	Sodium	43.10 mg
CALORIES	20.00 kcal	VB_2_	0.63 mg	Magnesium	13.00 mg
CARBOHYDRATES	4.10 g	Carotene	710.00 μg	Phosphorus	32.00 mg
PROTEIN	1.60 g	Riboflavin	0.13 mg	Iron	0.20 mg
FAT	0.30 g	Thiamine	0.03 mg	Zinc	0.29 mg
DIETARY FIBER	1.40 g	Niacin	0.40 mg	Copper	0.04 mg
VC	51.00 mg	Calcium	238.00 mg	Manganese	0.45 mg
VA	118.00 μg	Potassium	101.00 mg	Selenium	0.82 μg

## 2. Data Collection

All data presented in this review are summarized from the retrieved references, including scientific journals, book chapters, and dissertations. A comprehensive search was conducted across six academic databases, including PubMed, Embase, CNKI, Web of Science, Scopus, and the BOHAI University Library, in order to identify all relevant studies published up to 9 September 2025 (the final search date). These studies investigated the nutritional profile and bioactive components in radish leaves. The search strategy incorporated keywords related to nutrients and bioactive substances, such as “nutrients,” “metabolism,” “phytochemicals,” “carbohydrates,” and “fatty acids,” in combination with terms referring to radish leaves. The keyword was set as “radish leaves” for maximum relative references, with no other restrictions. Subsequently, references closely related to chemical compositions, traditional uses and pharmacological properties-including in vitro and in vivo investigations-were screened for further data extraction. The following types of studies were excluded: conference abstracts, cost–benefit analyses, editorials, conference papers, narrative literature reviews, systematic reviews, and meta-analyses. Additionally, backward citation searching was performed by reviewing the reference lists of the selected articles to identify any further relevant publications. This review analyzed a total of 500 research papers, with 96 of them being ultimately cited in the study.

## 3. Chemical Constituents

### 3.1. Alkaloids and Nitrogen Compounds

The structure of the carbohydrate moiety isolated from mature leaves consists of a continuous (1→3)-linked β-D-galactosyl main chain, a β-D-galactosyl residue with a (1→6)-linked β-D-galactosyl side chain, and an α-L-arabofuranyl residue linked in the outer region. This special structure endows it with excellent water retention and gelation properties, making it suitable as a natural food thickening agent for use in the food industry [16]. Two glycoproteins containing L-arabinose, D-galactose, L-fucose, 4-O-methyl-D-glucuronic acid, and D-glucuronic acid residues were isolated from brine extracts of mature radish leaves [17]. These acidic sugar groups provide certain ion-exchange capabilities and, as high-quality dietary fiber, also promote intestinal health [18]. It has been shown that most galactose residues are attached to a polypeptide backbone through a 3-O-D-galactosyl-serine linkage, indicating degradation of the glycoconjugates [4]. The arabinose 3,6-galactose found in radish leaves has been identified as similar to that present in hydroxyproline-rich protein, with unique sugar residues, including α-L-fucose-(1→2)-α-L-arabinose [19,20]. In addition, methylamine, pyrrolidine, and 1-(2′-pyrrolidinethio-3-yl-1,2,3,4-tetrahydro-beta-camparin-3-hydroxy acid have been detected in radish leaves [21]. These nitrogen-containing compounds possess antibacterial and antioxidant properties. Those components may have the potential to serve as natural preservatives, extending the shelf life of food [22]. Although significant progress has been made in the quantitative analysis of alkaloids and nitrogen-containing compounds, several key challenges persist in advancing future research, including the structural elucidation of these compounds, as well as comprehensive evaluations of their biological functions and potential synergistic interactions.

### 3.2. Enzymes

Plant enzymes have attracted much attention in the field of food science. Research has shown that cotyledons exhibit opposite developmental patterns of 1-O-acylglucose-dependent acyltransferase, 1-mustardylglucose: l-malate-mustardyltransferase (SMT), and 1-(hydroxycinnamoyl)glucose: 1-(hydroxycinnamoyl)glucose-hydroxycinnamoyltransferase (CGT) activities under light conditions [23]. Photocultivated seedlings showed high L-malate-mustardyltransferase and low 1-(hydroxycinnamoyl)glucose-hydroxycinnamoyltransferase activity, whereas dark-grown seedlings showed low L-malate-mustardyl acyltransferase and high 1-(hydroxycinnamoyl)glucose-hydroxycinnamoyltransferase activity [24]. These enzymes are involved in the synthesis of hydroxycinnamic acid derivatives. These components have significant antioxidant and antibacterial effects in food, which can extend the shelf life of the food and maintain the stability of its flavor [25]. Peroxidase and glutathione reductase activities in roots and leaves were significantly increased after 24 h of cadmium treatment, indicating that cadmium accumulation was positively correlated with peroxidase and glutathione reductase activities [24]. This positive correlation is of great significance in the context of food safety, given that cadmium—as a heavy metal—can enter the human body through the food chain. These enzymes not only have antioxidant properties but can also be used to detect heavy metal contamination in food, ensuring food safety [26]. PAGE analysis revealed the presence of multiple isoform variants of superoxide dismutase in the leaf tissue. Superoxide dismutase, as a natural antioxidant, can effectively inhibit lipid peroxidation and protect the components and flavors of food [27]. Waste radish leaves are rich in active components such as superoxide dismutase, peroxidase, and glutathione peroxidase, which not only have excellent antioxidant and antibacterial capabilities but can also be used as biological indicators to detect heavy metal pollution in food. As such, the extraction of compounds for the development of secondary products derived from radish leaves provides new ideas for food safety control.

### 3.3. Flavonoids

The flavonoid content in a radish leaf extract (RSL) was approximately twice as high as that in the root. Furthermore, the extract was subjected to high-performance thin-layer chromatography (HPTLC), which confirmed the presence of rutin-a polyphenolic bioflavonoid compound. This compound has been identified as a constituent of various plant parts that exhibits biological activity [28]. A number of studies have demonstrated the nutraceutical and pharmacological effects of this substance in the treatment of conditions such as cancer, cardiovascular disease and diabetes. Radish leaves may be a superior reservoir of naturally occurring bioactive substances, particularly those belonging to the flavonoid family, which are also present in the root [10]. Consequently, the dietary incorporation of radish leaves may result in an augmented intake of flavonoid compounds, given their high abundance [29,30].

In order to further clarify the specific composition and functions of flavonoids in radish leaves, a systematic analysis was conducted on the types, concentrations and biological activities of the main flavonoid substances in radish leaves. The detailed results are presented in Table 2. Table 2 not only demonstrates the main biological activities of rutin but also reveals the potential value of different flavonoid components in aspects such as antioxidation and anti-inflammation, further confirming the application value of radish leaves in functional foods.

### 3.4. Glucosinolates

Thiosulfate glucoside is an important secondary metabolite in cruciferous plants. Glucosinolate components in radish mainly include aliphatic glucosinolates and indolyl glucosinolates, which vary significantly in different radish materials and tissues; for example, the content of 3-indole methylglucosinolate glucosinolate is very low in seeds but is the dominant glucosinolate in leaves [37]. These indole-type glucosinolates are the precursors of their degradation product, indole-3-methanol, which has potential anticancer activity and can regulate hormone metabolism in the body and inhibit the growth of cancer cells [38].

Three glucosinolates (glucoside, glucoside, and 4-methoxyglucoside) were detected in radish leaves, and a total of nine glucosinolates were detected in radish seedlings (foraphanin, gluconolysase, glucoside, 4-hydroxyglucoside, glucosinolysin, glucoside, and 4-methoxyglucoside) [39].

The total aliphatic glucosinolate contents in the leaves of Xinlimei and super Zhengyan were similar, accounting for nearly 90% of the total glucosinolate content. The proportion of total aliphatic glucosinolates in leaves was only 80.59%. Of the aliphatic glucosinolates, 4-methyl-thio-3-butenyl glucosinolate accounted for more than 60% in the three varieties, making it the main aliphatic glucosinolate present in leaves [40]. Its degradation product is 4-methylthio-3-butenylisothiocyanate, which is the main source of the spicy flavor of radishes and directly affects the sensory quality of the food [41]. The content of 4-methylsulfoyl-3-butenylglucoside was relatively high, making it the second most important aliphatic glucoside component; furthermore, 4-methylsulfoyl-3-butenylglucoside is also a precursor of sulforaphane, which is currently recognized as an active substance with strong anticancer effects and, thus, has excellent prospects in the field of functional foods [42]. In addition, 2-allyl glucosinolate content in radish is relatively high, which is the secondary aliphatic glucosinolate of this variety. The total indole glucoside contents in the leaves of the three cultivars also showed significant differences. The total indole glucosinolate content in the leaves of Shawodin Radish was twice that in the leaves of Xinlimei and Super Zhengyan, while the indole-3-methyl glucosinolate content was the highest in Shawodin Radish, also about twice that determined in the other two varieties [43,44,45]. Radish leaves are rich in various glucosinolates such as 4-methyl-thio-3-butenyl glucosinolate, whose degradation product is 4-methylthio-3-butenylisothiocyanate, which improves the sensory quality of food. On the other hand, 4-methylsulfoyl-3-butenylglucoside has potential value in terms of functional foods due to its possible anticancer effects. Radish leaves have important potential for functional food development and medicinal applications, making them valuable resources for achieving the high-value utilization of agricultural by-products.

### 3.5. Organic Acids

A total of 23 fatty acids have been identified in the leaves of turnip, radish, and wild carrot. The predominant fatty acid in all samples was α-linolenic acid (C18:3n-3), a polyunsaturated fatty acid which possesses significant active functions and can be converted into DHA and EPA in the human body. It plays positive roles in preventing cardiovascular diseases such as coronary heart disease, as well as in the development of the nervous system [46]. This was followed by palmitic acid (C16:0), a saturated fatty acid which mainly serves to provide energy in food. Excessive intake can affect the metabolic health of the heart [47]. Turnip leaves were found to contain α-linolenic acid (C18:3n-3, polyunsaturated fatty acid), while radish and wild carrot leaves contained linoleic acid (C18:2n-6c, polyunsaturated fatty acids), an essential fatty acid for humans that can regulate lipid metabolism, promote the formation of cell membranes, and has anti-inflammatory and immune-regulating effects [48]. Notably, the radish leaves demonstrated the highest fatty acid content among the three samples, accompanied by the lowest polyunsaturated fatty acid content, accounting for 59%. Additionally, trace fatty acids—including cis-11-eicosenoic acid (C20:1) and eicosapentaenoic acid (C20:5n3)—have been identified in radish leaves [49], as well as caffeic acid and p-coumaric acid compounds [50]. The cotyledons of the seedlings were found to contain ferulic acid, erucic acid, isoascorbic acid, gentianic acid, hydrogenated cinnamic acid, p-hydroxybenzoic acid, salicylic acid, vanillic acid, and erucate and free malic acid, as well as linoleic acid, linolenic acid, malic acid, malonic acid, oleic acid, oxalic acid, and palmitic acid [7,51]. These compounds all possess certain biological activities; for instance, caffeic acid and p-coumaric acid have strong antioxidant properties and can inhibit lipid peroxidation, thereby extending the shelf life of food [52]. Meanwhile, salicylic acid and p-hydroxybenzoic acid also have certain preservative effects [53]. Although the leaves of radishes have generally been regarded as waste, they contain abundant organic acids, such as α-linolenic acid, linoleic acid, and caffeic acid, as active components with significant nutritional functions and great potential value in areas such as antioxidation and the prevention of cardiovascular diseases. The reasonable utilization of agricultural by-products, such as radish leaves, not only increases the utilization rate of functional foods but can also reduce the generation of agricultural waste.

### 3.6. Phenolic Compounds

The concentration of phenolic substances in radish leaves from Saudi Arabia was markedly higher than that observed in roots [54]. HPTLC analyses demonstrated that the radish leaf extract contained rutin, a flavanoid compound belonging to the polyphenolic category and representing one of the active components found in various plant parts. Rutin demonstrates quantitative antioxidant and antibacterial properties, capable of inhibiting lipid peroxidation and can be used as a natural preservative [55]. The observed differences in phenolic compound content among different plant parts may be attributed to a number of factors affecting polyphenol synthesis, including treatments applied after harvesting, intraspecific chemical diversity, the processes involved in plant breeding, and the influence of the surrounding environment.

The radish leaves contain ten phenolic substances (catechin hydrate, chlorogenic acid, erucinic acid, caffeic acid, epicatechin, para-coumaric acid, ferulic acid, trans-cinnamic acid, quercetin and kaferol) [44]. These phenolic substances possess excellent antioxidant properties and can extend the shelf life of food. For instance, caffeic acid and quercetin have been widely used in beverages [56]. The active components of radish leaves were caffeoyl malic acid, ferulic malic acid, p-coumaryl malic acid, and sinapin [57]. These phenolic substances exhibit excellent biological activity both during processing and application [58]. The content of phenolic acid in radish roots was much lower than that in leaves. The content of phenolic acid in radish roots was much lower than that in leaves. Furthermore, erucinic acid (1-glucosamine erucinic acid, malic acid and sucrose 6,3′-dicerucinic acid), The cotyledons of radish embryos were subjected to a process of extraction, whereby kaferol and unbound malic acid were separated from the remaining components. The radish leaves contain α and γ-tocopherol [59]. Vitamin E compounds are natural lipid-soluble antioxidants that are widely used in substances such as vegetable oils to prevent lipid oxidation [60].

### 3.7. Polysaccharides

The weakly acidic pectic polysaccharides and pectic acid from leaves was degraded by both exogenous and endogenous polygalacturonase, and that its galacturonic acid content (17.3–25.8%) was significantly lower than that of pectic acid, despite the neutral sugar composition of the two being essentially identical. The arabinogalactan-galactose side chains in pectin are longer or highly branched than those in pectin. The rhamnogalacturonan-I from radish leaves using pectinase and high-efficiency volumetric exclusion chromatography. REP-I consists mainly of galactans (Galactan, GalA, 22.2%), Rha (10.2%), Gal (32.6%) and Ara (27.5%). The repetitive structural unit comprises one main chain (→2)-Rhap-(1→4)-GalpA-(1→) and three side chains (α(1→5)Ara, β(1→4)GalA, and Ara-β-3,6-GalA), which branch at the C(O)4 position of the Rha residue of the main chain. The presence of a variety of polysaccharides in radish green leaves was confirmed, with good extraction potential [61]. The crude polysaccharide (Radish wide polysaccharide, RWP) isolated from radish leaves could be used as an immunostimulant and anti-metastatic agent. Furthermore, the analysis of the glycan composition revealed that RWP was a pectic polysaccharide. Furthermore, the structural characteristics of pectin polysaccharides enable them to form gels after processing, replacing fat and providing a texture similar to that of fat, which can be applied in the development of low-calorie foods [62]. Generally speaking, the leaves of radishes are regarded as waste materials. However, the pectin polysaccharides can be used as fat substitutes in the food industry, greatly increasing the value and development potential of agricultural waste materials.

### 3.8. Sufur Compound

Radish leaves contain only one bisulfate isomer of S-adenosylmethionine (AdoMet), which has a wide functional range within the biochemical context [63]. It functions as a metastable enzyme effector and precursor of spermine biosynthesis, as well as of spermine and ethylene [64]. Moreover, it functions as a methyl donor in the majority of physiological methylation reactions, whereby the donation of its carbon atom results in the conversion of AdoMet to a homocysteine analog [65]. The chemical and biochemical properties of AdoMet are largely attributable to its sulfide nature [63]. Additionally, sulfur-containing compounds were discovered, including 5-vinyl 2-oxazolidinethione, H_2_C=CHCH_2_NCS allyl-isothiocyanate, S-adenosylmethionine in the form of (S)-sulfanilamide, and CH_2_=CHCH_2_CH_2_C(SGLc)=NOSO_3_H potassium black mustard thiosulfonate [63]. These substances all play important roles. For instance, propenyl isothiocyanate is widely found in cruciferous vegetables and has strong antibacterial properties. It can be used as a natural preservative to inhibit foodborne pathogens [66]. These substances all play important roles. For instance, propenyl isothiocyanate is widely present in cruciferous vegetables and has strong antibacterial properties. It can be used as a natural preservative to inhibit foodborne pathogenic bacteria [66]; the sulfur-containing active components in radish leaves have significant antibacterial properties and can be used as a natural preservative. Radish leaves can be transformed from by-products into high-value functional food ingredients, providing a way to valorize agricultural waste.

### 3.9. Carotenoid

Carotenoids are known as light harvesters and are considered essential nutrients that the body is unable to produce. The color of these pigments is derived from carotenoids, which assist in the dispersal and pollination of plant seeds. Additionally, these pigments act as precursors to abscisic acid. A total of seven carotenoids have been identified in radish leaves, including purple xanthophyll, lutein, α-carotene, zeaxanthin, β-carotene, 9Z-β-carotene and 13Z-β-carotene. These carotenoids play significant roles in food. For instance, β-carotene, as a natural colorant, is widely used in beverages and margarine. It is also a precursor substance of vitamin A [67]. Additionally, lutein and zeaxanthin have a considerable impact on visual health and thus are often added to functional foods [68]. The synthesis of carotenoids occurs via endogenous precursors, which can only be obtained through dietary intake [39]. Furthermore, other carotenoids have been demonstrated to promote health and prevent disease. For example, research has demonstrated that carotenoids can mitigate immune system dysfunction, inhibit carcinogenesis, and prevent age-related degenerative diseases. Radish leaves, as agricultural waste, are rich in carotenoid active components. They can be used as ingredients in food processing and have functions such as nutritional enhancement and coloration. Therefore, through reasonable and effective extraction and utilization, the value of agricultural by-products like radish leaves can be enhanced, promoting the development of functional food industries.

Radish leave is a source of nutrients and phytochemicals, mainly includes alkaloids, nitrogen compounds, enzymes, flavonoids, glucosinolates, organic acids, phenolic compounds, sulfur compounds, carotenoid and ascorbic acid. The phytochemicals of Radish leave altogether are responsible for its biological activities. Figure 2 summarizes the various chemical constituents in radish leaves. Among them, the glucosinolates have gained enormous interest in the pharmaceutical industry, especially in the designing of anticancer and anti-inflammatory drugs.

## 4. Biological Activities

### 4.1. Acetylcholinesterase Inhibitory Activity

Acetylcholinesterase (AchE) is a pivotal enzyme in the biological transmission of neurotransmitters. Between cholinergic synapses, the enzyme degrades acetylcholine, thereby terminating the excitatory effect of neurotransmitters on the postsynaptic membrane and ensuring the normal transmission of neural signals in the organism [69]. Alzheimer’s disease (AD) is a neurodegenerative disease that is more common in the elderly. The most effective drugs for treating AD are acetylcholinesterase inhibitors, according to the cholinergic deficit hypothesis, which is generally accepted today.

A potent acetylcholinesterase (AChE) inhibitor was isolated from radish leaves. The active component was identified as cis-13-docosenamide. The study demonstrated that erucamide significantly inhibited AChE activity. Furthermore, the feeding and behavioral testing in mice demonstrated that erucamide prevented TMT-induced memory impairment. These findings indicate that radish leaves and their extract erucamide may be beneficial in preventing memory impairment associated with Alzheimer’s disease. It is postulated that this preventive effect may be achieved through modulation of cholinergic function. The research demonstrated that all six different extract fractions of radish leaves exhibited AChE inhibitory activity, with a discernible concentration-dependent relationship observed between their inhibitory activities and sample concentrations [9]. The order of the samples in terms of their AChE inhibitory activities was as follows: dichloromethane fraction, ethyl acetate fraction, petroleum ether fraction, n-butanol fraction, aqueous phase, and aqueous extract of radish leaves.

From the above results, the differences in the types of chemical components contained in different extracts directly affect the strength of their activities. The dichloromethane components and ethyl acetate components exhibit strong acetylcholinesterase inhibitory activity, which may be due to the presence of small molecule active substances such as amides and flavonoids, such as cis-13-icosamide. These substances can bind more easily with AcHE, resulting in a higher inhibitory intensity. While the components in the aqueous phase and the water extract of radish leaves mainly consist of polysaccharides and amino acids, their molecular structures and polarity have a weaker binding ability with AcHE, leading to a lower inhibitory intensity of their activities. However, the intensity of activity is not only determined by the different active components but also related to the relative content of active substances in the actual components. In the dichloromethane components and ethyl acetate components, the relative content of cis-13-icosamide may have reached or exceeded the concentration at which it can exert its effect, so the effect of expressing the intensity of activity is better. Therefore, the acetylcholinesterase inhibitory activity of radish leaves is the result of the combined action of specific types of chemical components and the small molecule active substances that exert their effects. This study provides a basis for the subsequent development of AcHE inhibitors from waste radish leaves.

### 4.2. Anti-Bolting Activity

A research team has predicted the existence of an anti-bolting compound that inhibited stem elongation synthesized under SD conditions (1), which was demonstrated to exert an anti-bolting effect [6]. The study established the absolute configuration of compound (**1**) at the C-2 position as (2S). Furthermore, an additional novel compound (**2**) and two previously characterized monoglycerol compounds (**3** and **4**) have been extracted from radish leaves [6]. Qualitative and quantitative analytical methods were developed for compounds **1**–**4**, utilizing two deuterium-labeled compounds (**8** and **9**) as internal standards. The distribution of these compounds was found to be more widespread in a variety of annual winter crops. The isolated compounds were also observed to inhibit the growth of Arabidopsis (*Arabidopsis thaliana*) seedling roots at concentrations of 25 and 50 micromoles [70]. However, the inhibitory effect of compound 1 was not dependent on the coronatine insensitive 1 (COI1) protein, which may indicate that additional signaling systems—i.e., beyond jasmonic acid signaling—are involved [71]. The above study conducted qualitative and quantitative analyses on compounds (**1**–**4**) isolated from radish leaves. It was found that the absolute configuration (2S) at the C-2 position of compound (**1**) might be closely related to its anti-bolting activity. The quantitative analysis further revealed that at concentrations of 25 and 50 micromoles, all compounds significantly inhibited the growth of the roots of Arabidopsis thaliana seedlings, indicating that the activity of the compounds strongly depends on their concentration. The above qualitative and quantitative analyses, as well as the discovery that there are other signaling systems besides the jasmonic acid signaling system, which are involved in exerting anti-bolting activity, provide a theoretical basis for the anti-bolting mechanism of the active components in radish leaves.

### 4.3. Anticancer Activity

A previous study demonstrated that a radish leaf extract inhibited the proliferation of all tested cancer cells—namely, A549, HepG2, MDA-MB-231, and MCF-7—using the MTT assay [54]. The viability of all treated cells was reduced in a dose-dependent manner, with the leaf extracts exhibiting more potent anticancer activity than the root extracts. The observed inhibitory effects on cell proliferation can be ascribed to the presence of a number of active ingredients in radish leaves, including various flavonoid compounds. Furthermore, another study found that administration of an ethanolic extract of radish leaves led to a notable reduction in MDA-MB-231 mammary gland carcinoma cells following a 48-h incubation period [72]. The extract demonstrated a pro-apoptotic effect on human colon cancer cells and also inhibited the metastasis of melanoma cells.

Zinc oxide nanoparticles (ZnO NPs) have been synthesized from radish leaves via a green synthesis process. The ZnO NPs were characterized using UV-Vis, FTIR, particle size analysis, scanning electron microscopy, and an X-ray diffractometer, and their anticancer activity was studied using A549 cells [73]. The UV-Vis spectra and particle size analysis confirmed the nanoscale nature of the prepared ZnO NPs. Furthermore, FTIR studies demonstrated the presence of various functional groups, while scanning electron microscopy and X-ray diffraction studies provided evidence that some crystalline spherical and fibrillar zincite crystals were present. The results of the cytotoxicity experiments demonstrated that the synthesized ZnO NPs exhibited enhanced cytotoxicity. This study was the first to investigate the synthesis of ZnO nanoparticles as anticancer agents using radish spp. Longipinnatus [74]. From the above results, it can be seen that the anticancer activities of extracts derived from radish leaves and synthetic nanoparticles are related to specific active substances and their doses. Various flavonoid compounds and other components that are rich in radish leaves exert anticancer effects by inhibiting cell proliferation, promoting the apoptosis of cancer cells, and inhibiting the metastasis of cancer cells. Additionally, further analysis revealed that the extract showed dose-dependent behavior, indicating that the relative content of active components directly affects the effectiveness of their action. Moreover, the ZnO NPs contained some crystalline spherical and fibrous zinc ore crystals and, in the cell toxicity experiment, it was shown that these significantly enhanced the cytotoxicity of A549 cells. This indicates that the active components in radish leaves can directly act on cancer cells or indirectly exert anticancer activities through nanomaterials.

### 4.4. Antihypertensive Activity

A recent study has provided updated insights into the effects of an ethyl acetate extract of radish leaves on high blood pressure in 11-week-old spontaneously hypertensive rats (SHRs) [75]. A downward trend in systolic blood pressure (SBP) was observed in the spontaneously hypertensive rats (SHRs) after administration of the radish leaf extract. The SBP in the normotensive and hypertensive control groups remained unchanged, while that in the group receiving 90 mg extract/kg body weight decreased from 214 mmHg to 166 mmHg—significantly lower than that of the control groups [14]. The results showed that the intervention did not affect angiotensin-converting enzyme expression in the serum, kidneys, or lungs. However, the extract was found to increase serum NO concentrations and enhance the activity of redox enzymes, including superoxide dismutase and thioredoxin reductase, in red blood cells (RBCs) [14]. There were no significant differences in the serum concentrations of Na(+) and K(+) among the groups, but fecal concentrations of sodium and potassium were increased. There were no statistically significant differences in the fecal sodium and potassium concentrations between the control subjects with normal blood pressure and those with hypertension; however, urinary sodium excretion was higher in healthy Wistar rats compared with the SHR, while no significant difference was observed in potassium excretion [14]. The study demonstrated that turnip leaf extract exerted hypotensive effects on SHRs by increasing serum nitric oxide (NO) concentrations, enhancing antioxidant enzyme activities (e.g., glutathione peroxidase and catalase) in erythrocytes, and elevating fecal sodium and potassium concentrations. The antihypertensive activity of the extract from radish leaves is the result of the combined action of multiple active components. Firstly, radish leaves are rich in various active components—such as polyphenols and flavonoids—which not only enhance antioxidant activity but also increase the concentration of NO in the serum, achieving the effect of lowering blood pressure through a multi-faceted synergy. On the other hand, specific active substances promote excretion at a certain dose, thus increasing the concentration of sodium and potassium in the feces; however, as there was no significant difference in the concentration of sodium and potassium in the blood, a dose of 90 mg/Kg is considered as an effective dose for the active components to exert their effects. These results indicate that the antihypertensive activity of the radish leaf extract is due to the combined action of multiple active components and various regulatory mechanisms, providing a scientific basis for its use in regulating hypertension.

### 4.5. Anti-Inflammatory Activity

The anti-inflammatory effects of radish leaf extract were experimentally assessed by Hye-Jin Park et al. [76]. The RSL (*Raphanus sativus* L.) powder was fractionated into n-hexane, dichloromethane, ethyl acetate, n-butanol, and water-soluble fractions. Initial screening was performed on LPS-stimulated RAW264.7 cells treated with each fraction. The chloroform fraction significantly inhibited nitric oxide release in these cells. Both mRNA and protein levels of inducible nitric oxide synthase were reduced in a dose-dependent manner, as determined by reverse dot blot and reverse transcriptase-quantitative polymerase chain reaction. Furthermore, the RSL chloroform fraction was found to reduce the expression of nuclear factor-κB (NF-κB), a key regulator of the transcriptional activation of inflammatory cytokine genes. The findings suggest that the RSL chloroform fraction inhibited endotoxin-induced production of cytokines and other pro-inflammatory compounds in RAW264.7 bone marrow-derived macrophages by modulating intracellular signaling pathways. The results show that the chloroform fraction of RSL inhibited the release of various substances involved in the initiation and perpetuation of inflammation, including inducible nitric oxide synthase (iNOS), cyclooxygenase-2 (COX-2), and pro-inflammatory cytokines. Additionally, RSL demonstrated anti-inflammatory effects in LPS-stimulated macrophages by inactivating NF-κB. Radish extract boosted a tolerance response by promoting the strongly transcription of the anti-inflammatory cytokine TGF-β and a trend upregulation of IL-10 [77]. The leaves of radishes are rich in various active components that exert anti-inflammatory effects. For instance, components such as glucosinolates, phenols, and isothiocyanates are present. The chloroform group will accumulate lipid-soluble phenolic compounds and the degradation products of glucosinolates, etc. These components can inhibit the expression of NF-κB, suppress iNOS, COX-2, and the release of pro-inflammatory cytokines. Moreover, the anti-inflammatory effect of the chloroform components is the best, indicating that the content of anti-inflammatory active components in the chloroform group is higher than that in other groups. In conclusion, the anti-inflammatory effect of radish leaves depends on the types and contents of active components in different components, providing an economical and effective option for managing and preventing inflammatory diseases.

### 4.6. Antimicrobial Activity

The caffeic acid exhibits antifungal activity against Sporothrix marcescens. It has been demonstrated to possess antibacterial and antifungal activity. Ferulic acid has been demonstrated to exhibit antimicrobial activity against a range of bacterial and fungal pathogens, including *Staphylococcus aureus*, *Bacillus subtilis*, *Corynebacterium*, diphtheria, *Aspergillus niger* and *Candida albicans*. These acids have been demonstrated to exert bacteriostatic activity against a range of bacterial species, including Gram-positive bacteria such as *Bacillus subtilis* and *Staphylococcus aureus*, as well as Gram-negative *Escherichia coli* and *Klebsiella pneumoniae*. Among these, p-hydroxybenzoic acids (hydroxycinnamic acid, p-hydroxybenzoic acid) have been shown to exhibit particularly pronounced inhibitory effects against Gram-positive bacteria [78]. Daikon leaves were enriched in flavonoids and glucosinolates, red radish leaves were high in phenolic acids. Two radish leaf extracts exhibited significant antibacterial activity against *K. pneumoniae* and *P. aeruginosa*. Daikon leaves prevented biofilm formation in *K. pneumoniae* by 74.2%. Additionally, the leaves of white icicle radish inhibited pyocyanin production in *P. aeruginosa* by 76.8%. Pearson’s correlation analyses revealed that the bioactivities were linked to various phenolic and sulfur compounds [79]. The antibacterial ability of radish leaves is closely related to the types and contents of their active components. There are significant differences in the active components among different varieties of radish leaves. For instance, the white radish leaves are rich in flavonoids and glucosinolates, while the red radish leaves are rich in phenolic acids. These compounds exert antibacterial effects through various mechanisms, such as disrupting the cell membrane structure of microorganisms and inhibiting the production of pyocyanin in *Pseudomonas aeruginosa*.

### 4.7. Anti-Obesity Activity

Green radish leaves contain a variety of polysaccharides with a favorable extraction effect, as demonstrated by their relative molecular mass ratio and monosaccharide composition analysis [80]. The polysaccharide components in radish green leaves can alleviate obesity caused by high-fat diets. Furthermore, it has been determined that their weight-loss activity may be associated with the maintenance of intestinal health and the alteration of lipid metabolism in adipose tissues. This can potentially be attributed to the probiotic abilities of polysaccharides, which drive intestinal microbial composition changes, improving gut barrier function, modulating the gut microbiota, and enhancing lipid metabolism as a result. Consequently, turnip greens may be a potential functional food source for the prevention of weight gain and obesity-related metabolic disorders.

The anti-obesity effects of turnip leaf extract (MU-C) and turnip leaf extract with 3% citric acid (MU-CA) have been experimentally assessed by Yun-Seong Lee et al. [81] in high-fat diet (HFD)-induced C57BL/6 mice. Additionally, the effect of turnip leaf extract on adipogenesis was investigated using 3T3-L1 adipocytes. The results demonstrated that the consumption of MU-C significantly reduced adipose weight, and histopathological analysis also confirmed that the size of adipose tissue was significantly reduced in mice treated with MU-C. This study provides a foundation for further investigation into the potential clinical application of MU-C as a drug for obesity prevention. In conclusion, the anti-obesity activity of radish leaves may be closely related to polysaccharides, due to their close connection to prebiotic activity. Therefore, by improving the intestinal barrier function, regulating the intestinal microbial flora, and enhancing lipid metabolism, the goal of reducing obesity can be achieved. On the other hand, when using two different extraction methods—namely, MU-C and MU-CA—the content, types, and proportions of effective active components in the extracts will vary. Experimental results indicate that the content, types, and proportions of the anti-obesity active components in radish leaves will affect their inhibitory effects on lipid production and, ultimately, their anti-obesity effects. This discovery provides a scientific basis for further developing functional foods or drugs targeting obesity, while helping reduce the waste due to discarding radish leaves.

### 4.8. Antioxidative Activity

Scavenging experiments using DPPH and ABTS free radicals, along with the H_2_O_2_-induced MRC-5 progeria model, revealed that a radish leaf methanol extract had antioxidant activity 1.8 times greater than that of vitamin C, with the main antioxidant substances identified as phenolic acid and flavonoids [82]. Noman et al. found that RSL showed dose-dependent DPPH and ABTS scavenging activity, as well as dose-dependent antioxidant activity and, in the DPPH scavenging activity test, RSL showed stronger antioxidant activity than radish root [54]. Goyeneche et al. has also experimentally verified that radish leaf extract has stronger antioxidant properties than roots. The antioxidant activities of the leaves and roots were found to be 39.48 mmol and 11.09 mmol TE/100 g d.m., respectively, according to an ORAC analysis. The most abundant free and bound phenolic compounds of roots and leaves were pyrogallol and vanillic acid, and epicatechin and coumaric acid, respectively [19]. Wu et al. found that radish flavonoids had a positive effect on both OH and O_2_, with obvious scavenging effects. Additionally, the scavenging effect increased with an increase in flavonoid concentration at the tested dose [83].

Luo et al. reported on the efficacy of a radish leaf extract in protecting against oxidative damage in human lung cellular models (MRC-5) [84]. The F2 fraction was obtained through the fractionation of radish leaves using a combination of polar solvents and an AB-8 macroporous resin column and was subsequently evaluated. Pretreatment with the F2 fraction before exposure to H_2_O_2_ considerably enhanced cell viability and intracellular antioxidant enzyme activity. Additionally, F2 pretreatment reduced malondialdehyde (MDA) levels, mitigated the increase in cytoplasmic reactive oxygen species (ROS) induced by H_2_O_2_, and restored the compromised mitochondrial membrane potential (MMP). Furthermore, F2 pretreatment downregulated the pro-apoptotic protein Bax and upregulated the anti-apoptotic protein Bcl-2, suggesting an initial protective mechanism. In conclusion, F2 in radish leaves can be employed as a conduit for antioxidants, thereby safeguarding the lungs from oxidative damage. Luo et al. investigated radish leaf as the research object and drosophila as the model organism to study the effects of different polar parts of a radish leaf ethanol extract on the organism’s lifespan and reproductive activity [81]. They reported that the total phenol and flavonoid contents and antioxidant activity were the highest in the ethyl acetate fraction.

The radical-scavenging capacity of radish leaves and buds is the primary source of their antioxidant activity [10,85], which has been demonstrated in laboratory and animal studies [86]. The antioxidant properties of radishes are attributed to their high concentrations of glucosinolates, anthocyanins, ascorbic acid, and polyphenols [87]. It can therefore be surmised that radish leaves may be instrumental in the prevention of diseases related to oxidative stress.

In conclusion, despite the substantial progress achieved in investigating the antioxidant activities of radish leaves, several critical areas warrant further in-depth research. First, owing to the considerable diversity among radish species, more precise identification and comprehensive analysis of bioactive constituents in different radish leaves, as well as the molecular mechanisms underlying the synergistic interactions among multiple components, remain to be elucidated. Second, the establishment of an optimal dosage regimen for radish leaf consumption to achieve maximal antioxidant efficacy has yet to be scientifically determined, which requires further investigation.

### 4.9. Antiulcer Activity

In a seminal study, Devaraj et al. investigated the antiulcer activity of radish leaf extract in two models of gastric ulceration: acetic acid-induced chronic gastric ulcer and pylorus ligation-induced gastric ulcer in rats [88]. The results of the acute oral toxicity studies demonstrated that the turnip leaf extract was safe at doses up to 2000 mg/kg per oral dose. Consequently, a dose one-tenth of this was selected for evaluation of the antiulcer activity. In the acetic acid-induced gastric ulcer model, the turnip leaf extract demonstrated a notable protective effect against acetic acid-induced ulcers in comparison to the control group. Furthermore, in the pylorus ligation-induced ulcer model, the extract exhibited a marked protective effect, as evidenced by a reduction in the ulcer index, total acidity, and free acidity.

Currently, the majority of studies investigating the antiulcer activity of radish leaves are limited to in vitro cell experiments or in vivo animal models, with the underlying mechanisms remaining largely unelucidated and a lack of high-quality clinical evidence. Future research on the anti-ulcer effects of radish leaves should focus on in-depth exploration of the structure–activity relationships and molecular mechanisms of various bioactive components, particularly their target sites, involved signaling pathways, and synergistic interactions within complex biological systems.

### 4.10. Immunological Properties

SON et al. found that crude polysaccharide isolated from radish leaves (RWP) can be used as an immune stimulant and anti-metastasis agent, demonstrating that peritoneal macrophages stimulated by RWP significantly increased the production of various cytokines [61]. In the determination of natural killer (NK) cell activity, oral administration of RWP could significantly enhance the cytotoxicity of NK cells against mouse lymphoma. In addition, pretreatment of NK cell inhibitors partially reduced the inhibitory effect of RWP on lung metastasis. The results showed that RWP has a strong immune-enhancing effect and good anti-metastasis activity, which are related to the activation of various immune factors. RWP has also been proven to have a strong anti-tumor metastasis effect through immunostimulating activity. REP-I isolated from RWP enhanced the secretion of interleukin-6, interleukin-12, and various cytokines (e.g., tumor necrosis factor) by peritoneal macrophages, which have strong immunostimulating effects.

Subsequently, in 2022, the research team elucidated the intracellular signaling pathway mechanism underlying the involvement of RG-I polysaccharide (REP-I) purified from radish leaves in macrophage activation [89]. REP-I primarily consists of the sugars GalA (22.2%), Gal (32.6%), Ara (27.5%), and Rha (10.2%). REP-I also reacted with β-glucosyl Yariv reagent (29.8%), suggesting the presence of arabino-β-3,6-galactan. Furthermore, methylated-product analysis revealed that REP-I contains 13 different glycosyl linkages, including 4-linked GalpA (21.0%), 2,4-linked Rhap (7.0%), 4-linked Galp (5.8%), 5-linked Araf (10.1%), and 3,6-linked Galp (7.9%), which are characteristic of RG-I. The addition of REP-I to macrophages resulted in enhanced gene transcription and secretion of immune-related mediators, including interleukin (IL)-6, tumor necrosis factor (TNF)-α, and nitrogen oxide (NO). In addition, Western blotting and immunocytochemical analysis demonstrated dose-dependent phosphorylation of mitogen-activated protein kinase (MAPK) and nuclear factor-κB (NF-κB) pathways by REP-I. A study utilizing a range of inhibitors demonstrated that the impact of REP-I on NO release was predominantly mediated by C-Jun N-terminal kinase (JNK) and NF-κB. It was confirmed that several PRRS and phosphorylated MAPK and NF-κB were closely related to the activation of REP-I macrophages isolated from radish leaves [89].

Immunity is a fundamental physiological mechanism in the human body, which serves to maintain health by defending against infections caused by foreign pathogens. It is broadly categorized into two types: innate immunity and adaptive immunity. The immune system not only provides defense against external threats but also accurately identifies and eliminates harmful substances within the body, thereby ensuring internal homeostasis. Additionally, it is capable of specifically recognizing mutated cells, thus playing a crucial role in preventing the development of cancer. Radish leaves exhibit notable immunomodulatory properties, with their sugar-derived compounds demonstrating potential to significantly enhance immune function. However, understanding of the mechanisms through which radish leaf polysaccharides interact with the immune system remains incomplete at present; for instance, the precise signaling pathways and mechanisms through which these polysaccharides promote the proliferation of T and B lymphocytes are not yet fully elucidated. Moreover, the impacts of radish leaf polysaccharides on nucleic acid and protein metabolism, as well as their interactions with various immune-related factors, require further investigation. Therefore, more in-depth studies are necessary to fully uncover their immunological effects and underlying molecular mechanisms.

### 4.11. Intestine Motility Stimulation

In a seminal study, Gilani and Ghayur (2004) demonstrated that a crude radish leaf extract obtained using Raphanus sativus leaves had dose-dependent spasmodic effects on the ileum and colon of guinea pigs. This effect was insensitive to atropine but could be completely eliminated by pyranamine, suggesting that it is related to histaminergic (H1) receptors. Petroleum t, chloroform, and water extracts showed histaminergic activity in ileum, with the water extract having a stronger effect. Studies have shown the presence of histaminergic components as well as weak spasmolytic agents, thus providing a good mechanistic basis for the plant’s traditional use for constipation [90]. On this basis, they further studied the irritant and relaxation properties of Raphanus sativus leaves using intestinal and uterine specimens of different species and performed activity-oriented classification of radish leaf extracts. The presence of saponins and alkaloids in the Raphanus sativus leaves was reported. They also induced spasmosis in rabbit jejunum, which was partially blocked by atropine, as well as in rat fundus and uterus. In contrast, it was found that Raphanus sativus leaves had no stimulative effect on the rat ileum but showed an inhibitory effect on ACh dose–response curves. Additionally, the aqueous portion—which contained strong saponins—was more effective in reducing spasticity than the nonpolar portion. A mild relaxant effect was also observed in the rabbit jejunum at lower doses (0.1–0.3 mg/mL) but not against K+-induced contractions, ruling out a calcium channel-blocking effect. The existence of species-dependent gastrointestinal effects of radish was partially mediated by cholinergic receptors in rabbit and rat tissues, but histaminergic activation was observed in guinea pigs, providing a scientific basis for its use in intestinal and uterine diseases [91].

Although radish leaf polysaccharides have demonstrated stimulatory effects on intestinal peristalsis, several limitations currently remain. Extraction and purification techniques for these polysaccharides have not yet reached the efficiency and scalability required for industrial production. Furthermore, the existing findings are primarily based on experimental studies using murine models, while clinical evidence from human population studies remains limited. This gap may be attributed to the labor-intensive and time-consuming nature of the extraction process, low yield, and constraints in experimental design. With advancements in overcoming production bottlenecks, research into the clinical applications of radish leaf polysaccharides is expected to expand. When combined with studies aimed at enhancing their stability and bioavailability, radish leaf polysaccharides hold promise for broader therapeutic use in the management of gastrointestinal and uterine disorders in the future.

### 4.12. Serological Activity

Tsumuraya et al. demonstrated that Arabinogalactan-proteins (AGPs) from radish leaves may be responsible for the expression of their serological activity [17]. AGPs consisted of consecutive (1→3)-inked β-D-galactosyl backbone chains having side chains of (1→6)-linked β-D-galactosyl residues, to which α-L-arabinofuranosyl residues were attached in the outer regions. In immunoreactivity with rabbit anti-radish leaf AGP antibodies, root AGPs were found to share antigenic determinants with seed and leaf AGP antibodies. Furthermore, arabinoxyl-3,6-galactose was associated with a hydroxyproline-rich protein fraction, which may determine the serological H-like activity of their AGPs.

Research on the serological activity of radish leaves is relatively outdated. Since the last investigation conducted in 1988, no subsequent studies have been carried out to further explore this phenomenon. The early findings regarding the serological activity of radish leaves have demonstrated significant potential for application in the medical field as well as in the high-value utilization of radish leaf resources. Therefore, it is of considerable importance to conduct in-depth investigations in future research to better understand and harness this property.

### 4.13. Other Activities

A novel heterogeneous catalyst was synthesized from unused radish leaves and used to prepare biodiesel from waste soybean oil (SWCO) and oblique algal oil (OSO). The synthesis catalysts were characterized using FTIR, SEM, and other methods, and the results showed that the highest SWCO and OSO conversion rates of 98.0% and 91.32% were obtained within 150 and 90 min at 60 °C, respectively. The catalyst exhibited excellent recyclability, achieved an oil conversion rate of 90.57% after four consecutive cycles, and produced biodiesel with fuel characteristics that meet international standards. The synthesized catalyst had the characteristics of high efficiency, cheapness, renewability, and green environmental protection, which can help to reduce the production cost of biodiesel. Waste radish leaves were used as raw materials to convert homologous alfalfa oil with high free fatty acids (FFAs) into biodiesel. The results showed that the physical and chemical properties of the biodiesel produced were within the scope of the biodiesel standard specification. It can be reasonably deduced that the two-step conversion method, which employs heat-treated radish leaves as a solid catalyst, has the potential to be a highly effective means of producing biodiesel from high FFA algal oil [92,93].

Radish leaf extract is a good biodegradable corrosion inhibitor. In particular, its efficiency in the corrosion inhibition of low carbon steel increases with increasing extract concentration but decreases with the increasing temperature. EIS, SEM, FTIR, and ultraviolet visible spectroscopic analysis showed that the corrosion inhibitor molecules were adsorbed onto the surface of low carbon steel to form a protective film, preventing corrosion caused by corrosive media [94,95]. The adsorption of RLE on the surface of mild steel follows the Langmuir adsorption isotherm model. Adsorption thermodynamic parameters indicated that physical adsorption and chemical adsorption occurred simultaneously. Theoretical calculations showed that folic acid and catechins have a greater corrosion inhibition effect than RLE.

Radish leaves exhibit a remarkable spectrum of biological activities, underpinning their potential value in health and industry. The mechanisms of action of various biological activities in the radish leaves are illustrated in Figure 3. They primarily demonstrate potent acetylcholinesterase (AChE) inhibitory activity, attributed to compounds, such as erucamide, which are found in specific solvent fractions (e.g., dichloromethane). This activity suggests potential for preventing the memory impairment linked to Alzheimer’s disease. Radish leaves also possess anti-bolting activity, containing compounds that can inhibit bolting and root growth via non-COI1 pathways. The various compounds present in radish leaves, especially the various flavonoids, exhibit significant anticancer activity. Furthermore, leaf-synthesized ZnO NPs enhance anticancer cytotoxicity. Antihypertensive activity has been observed in experiments in spontaneously hypertensive rats, with the extract from radish leaves having potential for the treatment of hypertension. In terms of anti-inflammatory activity, the chloroform extract of radish leaves can inhibit inflammation by suppressing NF-kB and increasing anti-inflammatory cytokines. The phenolic compounds and sulfur-containing compounds present in radish leaves exhibit broad-spectrum antibacterial activity against both bacterial and fungal pathogens. Furthermore, the anti-obesity biological activities of radish leaves are also important. Radish leaf polysaccharides can achieve anti-obesity effects by regulating the intestinal flora and lipid metabolism. Antioxidant activities are also an important biological aspect of radish leaves, which mainly demonstrate strong antioxidant effects through phenolic substances and flavonoids.

A comprehensive review of the literature focused on the biological activities of radish leaves indicates that, although numerous studies have confirmed their efficacy, existing investigations have predominantly focused on general biological effects rather than delving into the identification of specific active constituents and their underlying mechanisms of action, thus hampering the practical application of radish leaves. Furthermore, existing research has primarily relied on animal models and in vitro cell systems, with a notable absence of studies examining the in vivo effects in humans and determining the optimal dosage for intake. This gap in knowledge constrains our understanding of their real-world efficacy and limits their broader utilization in the fields of food science and pharmaceutical development.

Therefore, future research should prioritize the establishment of component–effect relationship profiles to precisely define the key bioactive substances responsible for the diverse biological functions of radish leaves. Additionally, advanced biotechnological approaches—including molecular biology, metabolomics, and proteomics—should be systematically employed to conduct more extensive in vitro, in vivo, and clinical studies. These efforts are expected to facilitate the identification of specific bioactive components and their molecular targets, thereby elucidating the compositional and mechanistic basis of the bioactivities of radish leaves. Such insights will provide a robust scientific foundation for the further development and application of radish leaves in both nutritional and medicinal contexts.

## 5. Conclusions and Future Prospects

As a common agricultural waste by-product, radish leaves have broad development prospects in the pharmaceutical and food industries. This study reported the structures of compounds that have been isolated from radish leaves and discussed their chemical and biological activities (see Figure 2 and Figure 3). These compounds can be grouped into structural categories, including carbohydrates, enzymes, flavonoids, glucosinolates, organic acids, phenolic compounds, sulfur compounds, polysaccharides, and other components. Their biological activities include antioxidant properties, acetylcholinesterase inhibitory activity, anti-convulsive activity, anticancer, antihypertensive, anti-inflammatory, antibacterial, anti-obesity, antiulcer, and intestinal motor stimulation, among others.

Numerous research groups have published extensive studies demonstrating that the phytochemicals present in radish leaves offer well-documented health benefits, particularly in promoting cardiovascular health, exhibiting potent anticancer activity against various tumor types, and significantly enhancing immune system function. These therapeutic effects are primarily attributed to bioactive constituents, such as polyphenols and flavonoids. Consequently, these compounds hold considerable application potential in human nutrition, serving not only as functional dietary supplements and nutraceuticals but also in the pharmaceutical and cosmetic industries. While radish leaves are not the sole natural source of flavonoids, polyphenols, and triterpenoids—many other plants possess similar bioactive profiles—their utilization represents a promising strategy for the high-value conversion of vegetable by-products and the reduction in agricultural waste. Therefore, further research and practical applications of phytochemicals derived from radish leaves are warranted.

Radishes are a rich source of nutrients and phytochemicals; particularly phenolic compounds, thioglucosides, flavonoids, and organic acids. Many of these phytochemicals are highly concentrated in the leaves and sprouts, which is especially crucial in the context of Western diets that prioritize root consumption. Additionally, radish leaves can enhance the flavor and nutritional value of dishes when used as an ingredient. Radish leaves may be considered a potential raw material for the development of nutraceuticals against infectious and non-infectious diseases. Although radish leaves are a traditionally discarded by-product, they can serve as a valuable natural source of nutrients and bioactive compounds with beneficial effects on human health. Furthermore, the compounds extracted from these underutilized plant parts can be employed for food preservation purposes, circumventing the utilization of synthetic additives and facilitating the development of novel natural ingredients.

A comprehensive review of domestic and international research on radish leaves indicated that their overall utilization remains low, resulting in both resource waste and environmental pollution. Despite notable advancements in the study of functional components derived from radish leaves, several challenges persist:(1)The concentrations of bioactive constituents in radish leaves are generally low and the associated extraction and purification processes are costly, thereby failing to meet the increasing demands of the market.(2)Although multiple bioactive compounds and their physiological functions have been identified, further research is required to determine which specific components are responsible for these effects and to elucidate their underlying mechanisms of action.(3)While numerous radish varieties exist, previous research has predominantly focused on white radish leaves, with limited investigations into the bioactive properties of other commonly consumed radish leaf types.(4)The majority of existing studies center on optimizing extraction methodologies, refining purification techniques, conducting structural characterization, and evaluating biological activities. Although these efforts have improved the extraction efficiency and compound yield to some extent, they have not effectively resolved the fundamental imbalance between supply and demand. Therefore, enhancing the yield of bioactive compounds while reducing production costs remains a critical issue requiring urgent attention.(5)The development of commercial products derived from radish leaves is still in its early stages, with a limited range of available formulations. Future research should prioritize the development of composite products incorporating radish leaves, which could help to mask their inherent bitterness while providing enhanced functional properties and facilitating their transformation into high-value-added applications.

Therefore, future research on radish leaves should focus on further improving extraction methodologies, elucidating their structural and conformational characteristics, and evaluating functional bioactivities. Such efforts are expected to facilitate the discovery of novel bioactive compounds, clarify the structure–activity relationships of functional constituents, and identify optimal extraction protocols which are suitable for industrial application. These advancements will provide essential theoretical and technical support for the development of innovative functional foods derived from radish leaves, better aligning them with consumer needs. Moreover, comprehensive identification of the composition, structure, and physicochemical properties of bioactive compounds in radish leaves through mass spectrometry is necessary to enhance the bioavailability of effective components and expand their potential therapeutic applications, particularly in the management of inflammation and disease prevention. It is recommended to align such research with market demands and intensify efforts toward the development of radish leaf-based nutraceuticals and functional food products.

The advancement of modern science and technology has created new opportunities for research and the application of radish leaves. Metabolomics enables comprehensive and dynamic analysis of the metabolic impacts of radish leaves, offering insights into the intrinsic relationships between their chemical constituents and biological activities. Artificial intelligence can be employed to predict bioactivities, optimize product formulations, and streamline manufacturing processes, thereby enhancing the efficiency and innovation of research. Furthermore, digital technologies can be applied to monitor and trace the entire production and processing chain of radish leaf-based products, ensuring quality control, traceability, and food safety.

In conclusion, radish leaves exhibit significant potential in terms of their bioactive components, physiological functions, and diverse applications. By integrating advanced technologies, conducting in-depth mechanistic studies, establishing robust quality control systems, and promoting product innovation, radish leaves are poised to play an increasingly important role in the fields of precision medicine, sustainable food production, and cosmeceuticals. Moreover, through the high-value utilization of vegetable by-products, radish leaves can contribute meaningfully to improving public health, driving socio-economic development, and supporting environmental sustainability.

## Figures and Tables

**Figure 1 foods-14-03270-f001:**
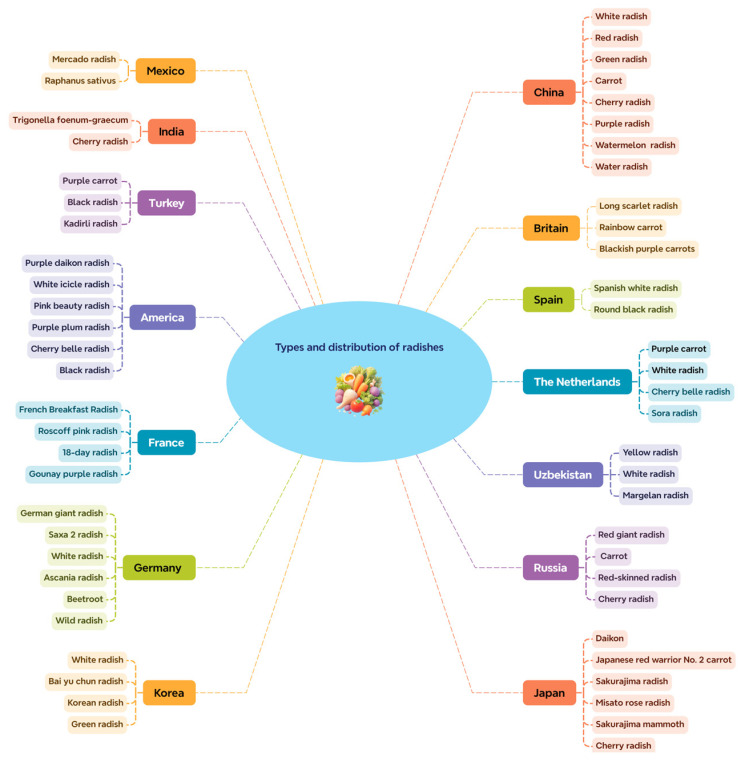
The varieties and distribution of radishes around the world.

**Figure 2 foods-14-03270-f002:**
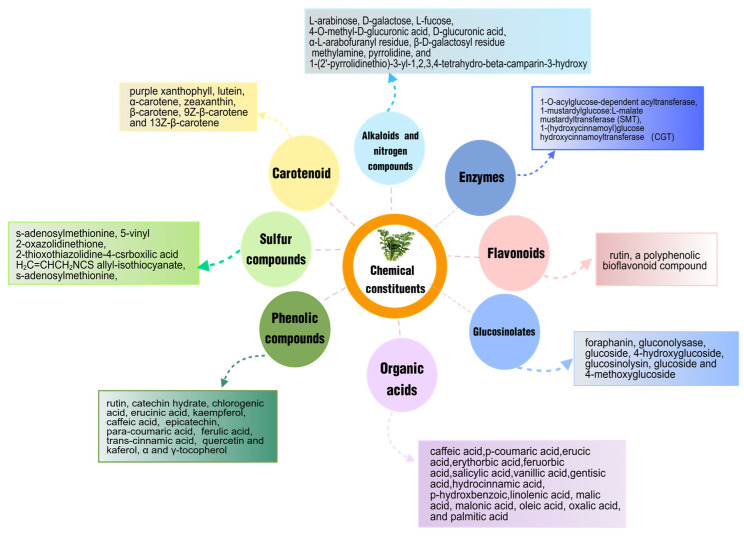
Various chemical constituents in radish leaves.

**Figure 3 foods-14-03270-f003:**
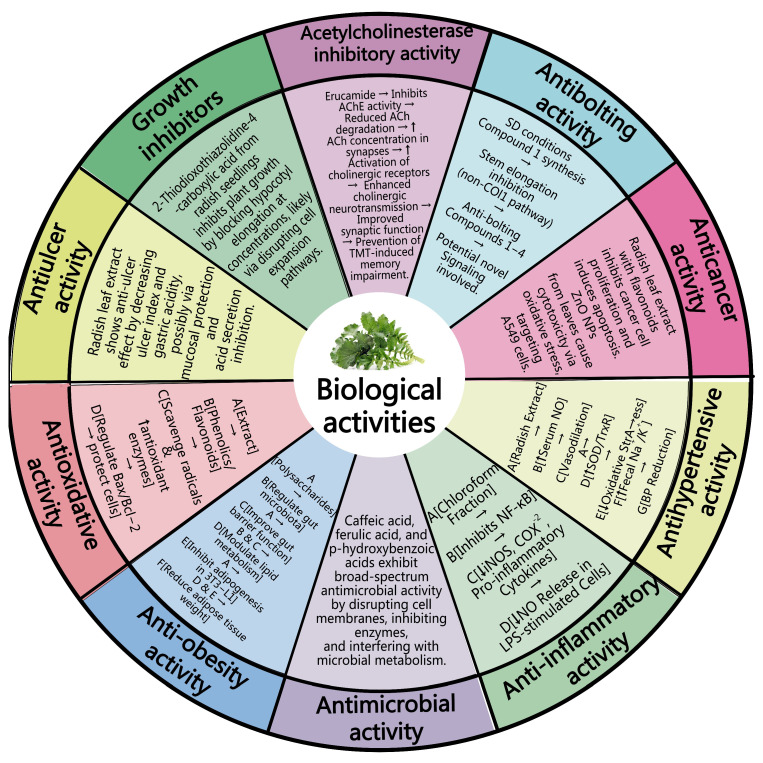
The mechanism of various biological activities in radish leaves. Arrows (→) indicate a causal relationship or the flow of biological events. The upward arrow (↑) indicates an increase in level or activation. The downward arrow (↓) indicates a decrease in level, expression, or inhibition.

**Table 2 foods-14-03270-t002:** Flavonoids in radish leaves and their biological activities.

Flavonoids	Concentration	Biological Activities
Rutin	The specific concentration was not mentioned.	Antioxidant: It can eliminate free radicals. Anti-inflammatory: It can inhibit the expression of genes, such as IL-6 and TNF-α, induced by lipopolysaccharides [31].
Quercetin	The specific concentration was not mentioned.	Quercetin has antioxidant and anti-inflammatory effects [31].
Sinapine thiocyabate	The specific concentration was not mentioned.	Studies have shown that it has significant antibacterial activity. At a concentration of 1 mg/mL, it showed a notable inhibitory effect on Escherichia coli and Staphylococcus aureus [32].
Kaempferol glycosides	The components extracted from the ethyl acetate fraction of radish leaves: 176.54 ± 1.02 mg/g dw.	Delays aging: This is achieved by reducing the activity of the aging-related β-galactosidase enzyme [33].
Total flavonoids	Radish leaves: 100.80 mg RE per 100 g dry weight [34].	Antioxidant activity: They can eliminate DPPH and ABTS free radicals and have the ability to reduce iron ions [35,36]. Anti-inflammatory activity: Inhibits the expression of genes related to inflammatory molecules [31]. Delays aging: Extends the average and maximum lifespan of fruit flies, increases the CAT/SOD activity in their bodies, reduces the activity of MDA, and demonstrates the potential to delay aging [33].

## Data Availability

No new data were created or analyzed in this study.

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
