# Peer review of "Nutritional and Phytochemical Characterization of Radish Leaves: A Comprehensive Overview"

_foods, 2025, doi:10.3390/foods14183270_

Round 1
Reviewer 1 Report
Comments and Suggestions for Authors
Line 31 - Error! Reference 31 source not found.. – Pay attention to errors when using reference manager
Improve your writing, some sentences don't make any sense and seem incomplete. For example, what did you mean by: "The extensive crossing and selection of radish (Line 36-37)"
What do you mean by “The content of radish is 3-10 times that of radish”
Table 1 is confusing, improve the way this data is presented
The purpose of the work after the composition table does not make sense to the text
In Topic 2.1, what is the importance of this composition?
The same for Topic 2.2, why the presence of this enzymes is important? They present any functional property?
Topic 2.3 – what other flavonoids are present? What is the concentration of these compounds? I suggest add a table with flavonoid composition, concentration and bioactivities associated with the compound.
What do you want to say in: “Glucosinolate is generally referred to as glucosinolate, which is formed by the enzyme action of endogenous glucosinolate”.
What is the importance of the different composition of glucosinolates? Discuss the data presented.
What does "Susss" mean?
The paper contains relevant information, but the authors failed to present the significance of this information. Overall, the data was presented in the manuscript, but no discussion was provided. Furthermore, the text includes several disconnected and contextually irrelevant sentences that compromise its clarity. I recommend a thorough revision to enhance coherence and flow.
Author Response
|
1.Point-by-point response to Comments and Suggestions for Authors Comments 1: Line 31 - Error! Reference 31 source not found. – Pay attention to errors when using reference manager. |
|
Response 1: Thank you for pointing this out. Reference 31 has been accurately cited and incorporated into the manuscript (Line 33).
|
|
Comments 2: Improve your writing, some sentences don't make any sense and seem incomplete. For example, what did you mean by: "The extensive crossing and selection of radish (Line 36-37)". |
|
Response 2: Thank you for your feedback. I have revised the sentence for clarity and completeness. Modify "This vegetable is a very important ingredient in many cuisines, especially Japanese, Chinese, and Korean. The extensive crossings and selection of radish. (Line 37-39)" to "This is largely due to the extensive crossing and selection of radish, particularly in East Asian countries, which have produced a large number of cultivars with an especially wide variation in root shapes. (Line 39-41)" |
|
Comments 3: What do you mean by “The content of radish is 3-10 times that of radish”. |
|
Response 3: Thank you for your feedback. Modify “Modern nutritional studies have shown that the nutritional value of radish leaves is higher than that of radish in many aspects, such as VC content is more than twice that of radish, calcium, magnesium, iron, zinc and riboflavin, folic acid, etc. The content of radish is 3-10 times that of radish [8].(Line 75-78)”to “According to modern nutritional studies, radish leaves exhibit higher nutritional value than radish roots in several aspects. For instance, the vitamin C content in radish leaves exceeds that in radish roots, by 2 times, while the levels of calcium, magnesium, iron, zinc, riboflavin, and folic acid were found to be 3 to 10 times higher in radish leaves compared to radish roots [10].(Line 116-120)” |
|
Comments 4: Table 1 is confusing, improve the way this data is presented.The purpose of the work after the composition table does not make sense to the text. |
|
Response 4: Thank you for your valuable suggestions. We have improved the presentation of the data in Table 1 and updated the relevant content in the manuscript accordingly. For detailed modifications, please refer to Table 1 and the introduction section in the revised manuscript . |
|
Comments 5: In Topic 2.1, what is the importance of this composition?The same for Topic 2.2, why the presence of this enzymes is important? They present any functional property? Topic 2.3 – what other flavonoids are present? What is the concentration of these compounds? I suggest add a table with flavonoid composition, concentration and bioactivities associated with the compound. |
|
Response 5: Thank you for your valuable suggestions. We have revised the manuscript to provide a clearer explanation of the significance of the components in Section 2.1 and the functional properties of the enzymes in Section 2.2. In accordance with your recommendation, we have also added a new table in Section 2.3 to summarize the flavonoid compounds, their concentrations, and associated biological activities. Detailed revisions can be found in the green-highlighted sections of Sections 3.1–3.3 (previously Sections 2.1–2.3) as well as in the newly added content. |
|
Comments 6: What do you want to say in: “Glucosinolate is generally referred to as glucosinolate, which is formed by the enzyme action of endogenous glucosinolate”.What is the importance of the different composition of glucosinolates? Discuss the data presented. |
|
Response 6: Thank you for these valuable contributions. We are sorry for any unclear sentences and have made the necessary revisions. Additionally, we have followed the suggestions and expanded the scope of the discussion to highlight the functional and biological significance of its components. For detailed information regarding the revisions, please refer to the green-highlighted sections in Section 3.4 (previously designated as Section 2.4). |
|
Comments 7: What does "Susss" mean? |
|
Response 7: Thank you for pointing this out. This was a spelling error and has been corrected to “Sufur compound”. |
|
Comments 8: The paper contains relevant information, but the authors failed to present the significance of this information. Overall, the data was presented in the manuscript, but no discussion was provided. |
|
Response 8: Thank you for your valuable revision suggestions. We have thoroughly revised the main content in accordance with your comments and have added comprehensive discussion sections to each relevant part of the manuscript. |
|
2. Response to Comments on the Quality of English Language |
|
Point 1: None. |
|
Response 1: None. |
|
3. Additional clarifications |
|
None. |

Reviewer 2 Report
Comments and Suggestions for Authors
The authors of this manuscript focus on the recently popular trend of using/applying parts of plants that have long been treated as waste. Review papers are an important part of scientific work, allowing for the collection of a wealth of interesting information on a given topic in one place. However, such papers must be written with great care. Avoid unnecessary repetition of information, which greatly reduces the quality of the work.
Some of the references used are quite old. Are you sure there is no possibility of citing more recent literature?
Please note that the topic at hand concerns radish leaves, and it is these that should be discussed in detail, not other plants.
The summary should also focus on the topic of this manuscript and not refer broadly to the plants mentioned earlier. There is no need for that.
Comments on the Quality of English Language
Please consider having your manuscript read by a native speaker to improve the quality of the language.
Author Response
1.Point-by-point response to Comments and Suggestions for Authors
|
Comments 1: The authors of this manuscript focus on the recently popular trend of using/applying parts of plants that have long been treated as waste. Review papers are an important part of scientific work, allowing for the collection of a wealth of interesting information on a given topic in one place. However, such papers must be written with great care. Avoid unnecessary repetition of information, which greatly reduces the quality of the work. |
|
Response 1: Thank you for your valuable suggestions. We have conducted comprehensive revisions to the manuscript, systematically addressing and eliminating any unnecessary repetitions of information. |
|
Comments 2: Some of the references used are quite old. Are you sure there is no possibility of citing more recent literature? |
|
Response 2: Thank you for your suggestions.We have endeavored to update the references; however, due to the limited availability of recent research on radish leaves, some of the citations in our manuscript remain from older publications. |
|
Comments 3: Please note that the topic at hand concerns radish leaves, and it is these that should be discussed in detail, not other plants. The summary should also focus on the topic of this manuscript and not refer broadly to the plants mentioned earlier. There is no need for that. |
|
Response 3: Thank you for your suggestions. We have made the following modifications. 1. Abstract (L11–17):Modify “Vegetable consumption has been proven to reduce the incidence of numerous diseases. To meet consumer demands, researchers and industries are increasingly exploring innovative technologies to recover high-value compounds from food waste. The recycling of biowastes holds significant importance for pharmaceutical, medical, and food industries. Radish is a useful vegetable with diverse features and worldwide distribution.” to “Radish is a root vegetable that is widely consumed globally. Radish leaves are typically not consumed and regarded as by-products in agricultural, industrial, and domestic settings. Accumulating evidence suggests that radish leaves possess higher nutritional value compared to the roots, primarily due to their elevated levels of protein, ash, dietary fiber, and ascorbic acid. In light of the growing emphasis on waste reduction and value-added utilization, the application of radish by-products has garnered increasing attention.” 2. The content regarding radish leaves has been added to the introduction section and the conclusions and future prospects section. Detailed revisions can be found in the red-highlighted sections o as well as in the newly added content. 3. 4.10. Growth Inhibitors (originally Section 3.10) have been removed from the manuscript, as they are not relevant to the discussion on radish leaves. |
|
2. Response to Comments on the Quality of English Language |
|
Point 1: Please consider having your manuscript read by a native speaker to improve the quality of the language. |
|
Response 1: Thank you for your suggestions. To enhance the linguistic quality of the manuscript, we have made use of the professional language services offered by MDPI expert editorial team. |
|
3. Additional clarifications |
|
None. |

Reviewer 3 Report
Comments and Suggestions for Authors
Reviewer’s Report
To Authors:
A manuscript entitled “Nutritional and Phytochemical Characterization of Radish Leaves: A Comprehensive Overview” was submitted for publication in Foods.
In this report, the authors have reviewed the scientific literature on the radish leaves’ nutritional and phytochemical constituents. The authors have inserted a table with information about the nutritional content of leaves. They have also included (phyto)chemical constituents in the review, such as: alkaloids and nitrogen compounds, enzymes, phenolic compounds (flavonoids), organic acids, carotenoids, sulfur-containing compounds, and polysaccharides. In addition, they have turned attention to the biological activities of the leaves. acetylcholinesterase inhibitory activity, antibolting, anticancer, antihypertensive, antioxidative activities, immunological properties, etc.
The headings help readers navigate the text. The information and topic are very interesting and it seems correctly interpreted in general. Nevertheless, I would rather describe this report as a compilation of past papers (conventional review) than as a thorough and critical review of the literature.
In general, I would expect an in-depth and critical comment and assessment of the published literature. Authors should keep in mind that a good review sets the trend and direction of future research on the subject matter being reviewed. The readers should be inspired after reading your review. What must they do in this scientific field?
A review is not the author’s own work but is a representation from the author(s) point of view of recent high-quality work in an important field and is intended to provide an all-encompassing and in-depth presentation of the recent impactful developments, the opportunities, the failures, the challenges, the interfaces with other disciplines (and how these interfaces affect the science) in the field.
Given that you will not have space to review every paper in the literature, you should then explain your reasons for selecting certain papers. Your 'results' are your findings drawn from analyzing the literature. Finally, for your review to have a real purpose, you must state your conclusions and what implications they have for further research in your field. This is done in minor in this review report.
The key to the review of the literature is not to provide a shopping list of past papers, compounds, activities, etc.
A good literature review report generally answers the following questions (at least):
- What are the achievements and limitations of these recent/previous works?
- What gap do these limitations reveal?
- How does my/other work intend to fill this gap?
The author should answer these questions: why do we need this review on this topic, and what more will this review give us when we read it? What more is this review than previously published, e.g., https://www.sciencedirect.com/science/article/pii/S0924224421003058; https://www.sciencedirect.com/topics/agricultural-and-biological-sciences/radish.
The author should try to include some introductory and finalizing sentences in each section and sub-section of the manuscript.
The authors may think about the introduction of the Materials and methods section in the manuscript where information about: how many papers are evaluated, where they are collected – Scopus, WoS, Google Scholar, etc.; the year of analysed papers (since 2000 to 2025, etc.), and all other information which will be useful for the reader.
Although the topic of this review is welcome, I consider that the paper is insufficient for publication in its present form.
Specific comments and suggestions to the author are described below:
Introduction: In general, not a real introduction. It would be useful to include more numerical data for the radish as a crop: production amount, how many sorts and varieties are cultivated, and numerical data for the by-products (waste) of this plant. In addition, some morphometric data for the radish would also be useful with some picture material as supplementary materials. What’s the % of leaves of the whole plant? The Introduction is not a real background of the study, and the argumentation needs to be reinforced. It would be a good idea to compare roots and leaves and to discuss differences. Is there more investigation of leaves, or it still need studies? Additionally, the published paper analysis should also be useful. How many papers are published about chemical composition, about morphology, about application, etc.?
L78 – the sentence is not fully clear;
L80 – Surprisingly, the carrot leaves were introduced. Why, since the authors should focus on radish leaves?
Table 1 should contain a column with a reference cited for different nutrients. It would be useful to have the nutrients ordered by class.
Section 2. – The data should be organized as a table, and the text should contain critical comments and analysis of the data present in the table.
L202-204 – the sentences are duplicated.
L209 – The authors have written about pectic polysaccharides, but the sub-section is about phenolic compounds – introduce a new sub-section.
L223-224 – ‘Susss’ - !? Please, correct this title.
L263 – The sentence is not very clear.
L520 – This sounds like a conclusion.
Part 3 – Biological activity is interesting, but different activities should be commented keeping in mind the qualitative and quantitative composition of the radish leaves. A comparison with other proven compounds/plants would be good as well. Now, it was just described/retold studies found.
I would suggest the addition of text in the conclusion section or in the main body of the manuscripts, with future research needed and limitations of the different studies.
Author Response
|
1. Point-by-point response to Comments and Suggestions for Authors Comments 1: A manuscript entitled “Nutritional and Phytochemical Characterization of Radish Leaves: A Comprehensive Overview” was submitted for publication in Foods. In this report, the authors have reviewed the scientific literature on the radish leaves’ nutritional and phytochemical constituents. The authors have inserted a table with information about the nutritional content of leaves. They have also included (phyto)chemical constituents in the review, such as: alkaloids and nitrogen compounds, enzymes, phenolic compounds (flavonoids), organic acids, carotenoids, sulfur-containing compounds, and polysaccharides. In addition, they have turned attention to the biological activities of the leaves. acetylcholinesterase inhibitory activity, antibolting, anticancer, antihypertensive, antioxidative activities, immunological properties, etc. The headings help readers navigate the text. The information and topic are very interesting and it seems correctly interpreted in general. Nevertheless, I would rather describe this report as a compilation of past papers (conventional review) than as a thorough and critical review of the literature. In general, I would expect an in-depth and critical comment and assessment of the published literature. Authors should keep in mind that a good review sets the trend and direction of future research on the subject matter being reviewed. The readers should be inspired after reading your review. What must they do in this scientific field? A review is not the author’s own work but is a representation from the author(s) point of view of recent high-quality work in an important field and is intended to provide an all-encompassing and in-depth presentation of the recent impactful developments, the opportunities, the failures, the challenges, the interfaces with other disciplines (and how these interfaces affect the science) in the field. Given that you will not have space to review every paper in the literature, you should then explain your reasons for selecting certain papers. Your 'results' are your findings drawn from analyzing the literature. Finally, for your review to have a real purpose, you must state your conclusions and what implications they have for further research in your field. This is done in minor in this review report. The key to the review of the literature is not to provide a shopping list of past papers, compounds, activities, etc. A good literature review report generally answers the following questions (at least): 1. What are the achievements and limitations of these recent/previous works? 2. What gap do these limitations reveal? 3. How does my/other work intend to fill this gap? The author should answer these questions: why do we need this review on this topic, and what more will this review give us when we read it? What more is this review than previously published, e.g., https://www.sciencedirect.com/science/article/pii/S0924224421003058; https://www.sciencedirect.com/topics/agricultural-and-biological-sciences/radish. |
|
Response 1: We sincerely appreciate your valuable comments and suggestions. In response to your feedback, we have conducted a comprehensive revision of the entire manuscript, with particular emphasis on improving the introduction and conclusion sections. |
|
Comments 2: The author should try to include some introductory and finalizing sentences in each section and sub-section of the manuscript. |
|
Response 2: We sincerely appreciate your valuable comments and suggestions. In response to your feedback, we have conducted a comprehensive revision of the entire manuscript, with particular emphasis on improving Part 3 and Part 4. |
|
Comments 3: The authors may think about the introduction of the Materials and methods section in the manuscript where information about: how many papers are evaluated, where they are collected – Scopus, WoS, Google Scholar, etc.; the year of analysed papers (since 2000 to 2025, etc.), and all other information which will be useful for the reader. |
|
Response 3: Thank you for your suggestions. The following content has been added (Line 181-199). 2. Data Collections All data presented in this review are summarized from the retrieved references, including scientific journals, book chapters, and dissertations. A comprehensive search was conducted across six academic databases, including PubMed, Embase, CNKI, Web of Science, Scopus, and the BOHAI University Library, in order to identify all relevant studies published up to September 9, 2025 (the final search date). These studies investigated the nutritional profile and bioactive components in radish leaves. The search strategy incorporated keywords related to nutrients and bioactive substances, such as "nutrients," "metabolism," "phytochemicals," "carbohydrates," and "fatty acids," in combination with terms referring to radish leaves. The keyword was set as "radish leaves" for maximum relative references, with no other restrictions. Subsequently, references closely related to chemical compositions, traditional uses and pharmacological properties-including in vitro and in vivo investigations-were screened for further data extraction.The following types of studies were excluded: conference abstracts, cost-benefit analyses, editorials, conference papers, narrative literature reviews, systematic reviews, and meta-analyses. Additionally, backward citation searching was performed by reviewing the reference lists of the selected articles to identify any further relevant publications.This review analyzed a total of 500 research papers, with 96 of them being ultimately cited in the study. |
|
Response 4: Thank you for your suggestions. We added the following content to the Introduction. Line 56-60: The radish varieties cultivated in China are predominantly characterized by white skin and white flesh, as well as green skin and green flesh. Among a total of 189 recorded varieties, 72 exhibit white skin and white flesh, 50 exhibit green skin and green flesh, and 38 exhibit red skin and white flesh. The remaining types occur in relatively smaller numbers [5].
Line 71-102: According to the Food and Agriculture Organization of the United Nations (FAO), the global radish harvest area has exhibited a fluctuating downward trend in recent years, decreasing from 1,169.4 thousand hectares in 2010 to 1,085 thousand hectares in 2018. From 2018 to 2021, the area remained relatively stable at approximately 1,100 thousand hectares, with a slight recovery to 1,096 thousand hectares in 2021 compared to the previous year. In terms of global radish production, although the harvested area has generally declined, advancements in agricultural technology have led to increased yields per unit area, contributing to a steady upward trend in total production. Statistical data indicate that global radish production rose from 34.964 million metric tons in 2010 to 41.667 million metric tons in 2021. Regarding the geographical distribution of production, China remains the world's largest radish producer, accounting for 43.6% of global output in 2021. Other significant producers include Uzbekistan, the United States, Russia, and Germany, with annual outputs of 3.156 million, 1.433 million, 1.303 million, and 0.962 million metric tons, respectively. Taking China as an example, a rough estimate shows that the annual production of root and tuber vegetable waste is approximately 390 million tons. Radishes typically reach a height of 20 to 100 centimeters. The fleshy roots are generally elongated-oval, spherical, or conical in shape, and exhibit a periderm coloration ranging from green to white or red. The stems are branched, glabrous, and possess a slight powdery bloom. The basal and lower stem leaves of radish are characteristically large and exhibit a pinnately semi-lobed morphology. Radish leaves are clustered on the shortened stem during the growth period, and their morphological characteristics—including shape, size, and color—vary depending on the variety. Supplementary Material 1 provides diagrams of the roots and leaves of three common types of radishes. The leaf proportion also differs among radish varieties, ranging from 8 to 30 cm in length and 3 to 5 cm in width. The terminal lobe is ovate in shape, while the 4 to 6 pairs of lateral lobes are oblong, bearing blunt teeth and sparsely covered with coarse hairs. The upper leaves are oblong in form, displaying serrated margins or approaching an entire margin [8]. Analysis of radish leaf morphology indicated that, among 110 radish varieties with documented leaf type information, pinnately lobed leaf varieties are more prevalent, accounting for 73 varieties, whereas flat leaf varieties are comparatively fewer, comprising only 37 varieties [5].
Line 121-123: As shown in the food nutrition content table, the calcium content in radish leaves was found to reach 238 mg, which is the highest of all components. The content of VC in the leaves was 51 mg/100 g, which is twice the content in the root.
Line 144-178: In recent years, scholars at home and abroad have conducted comprehensive research on the utilization of radish leaves. Wu Haiqing et al. made pickles from radish leaves and systematically analyzed the changes in nutritional components and microbial dynamics at different processing stages [12]. Dong Zhouyong et al. used ultrasonic-assisted extraction to separate chlorophyll from radish leaves, with an average extraction rate of 0.413% [13]. Da-Hee Chung et al. demonstrated that an ethyl acetate extract of radish leaves has antioxidant and anticancer properties and showed the potential to regulate blood pressure in spontaneously hypertensive mice [14]. Radish leaves are a widely available resource. With the development of biology, nutrition and immunology, research on their bioactive components has also deepened. These bioactive compounds have been explored for various uses, including as functional ingredients in health products or food formulas with antioxidant, immunomodulatory and anti-obesity properties. In addition, due to their biological functions, radish leaf extracts may also serve as carriers for anticancer and hypoglycemic drugs. Out of 500 references, 96 met our inclusion criteria. Arough count shows that over 460 papers have explored the chemical composition, functional properties and application potential of radish leaves, while more than 40 studies have focused on their morphological characteristics. Despite some progress, several challenges remain: (1) research has mainly focused on white radish leaves, with less attention paid to other radish varieties; (2) there are differences in the extraction efficiency of functional components, and most extraction methods have not yet been industrialized; (3) insufficient research on product development limits the ability to meet the growing demands of the health food industry. Due to the limited research, the overall utilization rate of radish leaf by-products remains very low, leading not only to resource waste but also to environmental pollution. To address this issue, future research should focus on optimizing extraction techniques, clarifying the structural and conformational characteristics of bioactive compounds and evaluating their functional activities. These efforts are expected to help in discovering new bioactive substances, elucidating structure–activity relationship, and establishing optimal industrial-scale extraction schemes. Such research will provide a solid theoretical and technical foundation for the development of innovative radish leaf functional foods, thereby better aligning product development with consumer demands. In view of the above, this study presents a comprehensive review of the phytochemical and pharmacological effects of radish leaves based on the available research data, with a view to providing a basis and application prospect for further research on radish leaves. Supplementary Material 1 provides diagrams of the roots and leaves of three common types of radishes.
|
|
Comments 5: L78 – the sentence is not fully clear; |
|
Response 5: Thank you for the valuable revisions you provided. I have incorporated all the suggested changes into the manuscript accordingly. it is modified as: Line 116-120:According to modern nutritional studies, radish leaves exhibit higher nutritional value than radish roots in several aspects. For instance, the vitamin C content in radish leaves exceeds that in radish roots, by 2 times, while the levels of calcium, magnesium, iron, zinc, riboflavin, and folic acid were found to be 3 to 10 times higher in radish leaves compared to radish roots [10]. |
|
Comments 6: L80 – Surprisingly, the carrot leaves were introduced. Why, since the authors should focus on radish leaves? |
|
Response 6: Thank you for your suggestions. The section detailing the nutritional composition of carrot leaves has been removed from the manuscript. |
|
Comments 7: Table 1 should contain a column with a reference cited for different nutrients. It would be useful to have the nutrients ordered by class. |
|
Response 7: Thank you for your valuable suggestions. Following your recommendation, we have reorganized Table 1. With regard to the comment that "Table 1 should contain a column listing references for different nutrients," we would like to clarify that all data in Table 1 were sourced from a single research article. Therefore, we have included the reference in the table caption for clarity and consistency. |
|
Comments 8: Section 2. – The data should be organized as a table, and the text should contain critical comments and analysis of the data present in the table. |
|
Response 8: Section 3. – (Section 2 of the original manuscript) We sincerely appreciate your valuable suggestions. With regard to the comment that "the data should be organized in a table, and the text should include critical comments and analysis of the data presented in the table," we did attempt to present the data in tabular form during the initial drafting of the manuscript. However, due to the complexity of the content, we found that a table format could potentially lead to confusion and reduce readability for the audience. As a result, we opted for a graphical representation instead. Given the current time constraints, we have retained the graphical format in this version. Nevertheless, we fully recognize the merit of your suggestion and will certainly consider its implementation in future other review papers. |
|
Comments 9: L202-204 – the sentences are duplicated. |
|
Response 9: Thank you for your suggestions. The repeated part “The content of phenolic acid in radish roots was much lower than that in leaves.” has been deleted from the manuscript. |
|
Comments 10: L209 – The authors have written about pectic polysaccharides, but the sub-section is about phenolic compounds – introduce a new sub-section. |
|
Response 10: L370 – Thank you for your revision suggestions. The new subsection (3.7. Polysaccharides) has been successfully added to the manuscript. |
|
Comments 11: L223-224 – ‘Susss’ - !? Please, correct this title. |
|
Response 11: Thank you for pointing this out. This was a spelling error and has been corrected to “Sufur compound”. |
|
Comments 12: L263 – The sentence is not very clear. |
|
Response 12: L451 –452 Thank you for your revision suggestions. The manuscript has been updated to replace " A potent AchE inhibitor isolated from radish leaves. The active component was identified as cis-13-docosenamide (cis-13-docosenamide) by a graded isolation of the radish leaf extract. “with "A potent acetylcholinesterase (AChE) inhibitor was isolated from radish leaves. The active component was identified as cis-13-docosenamide.” |
|
Comments 13: L520 – This sounds like a conclusion. |
|
Response 13: Thank you for your revision suggestions. This section aims to offer a comprehensive summary and discussion of the fourth section (the third part of the original manuscript). |
|
Comments 14: Part 3 – Biological activity is interesting, but different activities should be commented keeping in mind the qualitative and quantitative composition of the radish leaves. A comparison with other proven compounds/plants would be good as well. Now, it was just described/retold studies found. |
|
Response 14: Thank you for your valuable suggestions. In response to your comments, we have expanded the discussion in Section 4 (originally Part 3 of the manuscript) to include perspectives on both qualitative and quantitative compositional aspects, recent advances in biological activity research, existing limitations, and potential future research directions. |
|
Comments 15: I would suggest the addition of text in the conclusion section or in the main body of the manuscripts, with future research needed and limitations of the different studies. |
|
Response 15: Thank you for your revision suggestions. Within Section "5. Conclusions and future prospects" of the manuscript, we have incorporated a discussion on the necessary directions for future research as well as the limitations associated with the various studies reviewed. |
2. Response to Comments on the Quality of English Language
|
Point 1: None. |
|
Response 1: None. |
|
3. Additional clarifications |
|
None. |

Round 2
Reviewer 2 Report
Comments and Suggestions for Authors
I do not have no further comments.
Reviewer 3 Report
Comments and Suggestions for Authors
A revision of the manuscript entitled “Nutritional and Phytochemical Characterization of Radish Leaves: A Comprehensive Overview” was resubmitted for consideration for publication in Foods. The authors have answered the reviewer’s questions and comments. I am pleased to note that the authors have taken into account a lot of notes and suggestions of reviewer(s) during revision and have made adequate adjustments and corrections in the manuscript. Now, the quality of the manuscript has improved significantly.
I consider that the paper would be of interest and is acceptable for publication in this form.